# Building the Framework for Sustainable Tourism in Príncipe Island

Francisco Silva [1,2,*] and Miguel Roque [3]

1   Centre of Geographical Studies, Instituto de Geografia e Ordenamento do Território (IGOT),
    University of Lisbon, 1600-276 Lisbon, Portugal
2   Centre for Tourism Research, Development and Innovation—CiTUR, Escola Superior de Hotelaria e Turismo
    do Estoril, 2769-510 Estoril, Portugal
3   Escola Superior de Hotelaria e Turismo do Estoril, 2769-510 Estoril, Portugal;
    miguel.roque.12263@alunos.eshte.pt
*   Correspondence: francisco.silva@eshte.pt

**Abstract:** Like many other Small Island Developing States, São Tomé and Príncipe's economy faces major weaknesses and constraints. These challenges are especially marked on the island of Príncipe, owing to its small size and double insularity. In recent decades, tourism, driven by international investment, has become a strategic sector for territorial development. This study assesses the suitability of this exogenous model and explores the feasibility of adopting a progressive change to a more community-centered tourism development model. Extensive fieldwork and multi-stakeholder collaboration have highlighted the need for a holistic, multi-dimensional strategy to secure this change. Such a strategy would prioritize local skills enhancement, infrastructure improvement, better governance and the diversification of tourism products and experiences. The gradual shift to a more endogenous approach in tourism development aims to strengthen sustainability across its multiple dimensions, ensuring more substantial and direct benefits for the local community and adding value to tourist services and experiences.

**Keywords:** small island development states; sustainable tourism; designing tourist experiences; Príncipe Island

## 1. Introduction

The Small Island Developing States (SIDSs) are recognized by the United Nations as a special group, sharing common challenges. The Earth Summit, in 1992, highlighted the need for specific strategies for SIDS, emphasizing their resource limitations, isolation from markets, and economic disadvantages [1]. The weakness and low competitiveness of the SIDS economies are the result of multiple factors, in particular small market size, limited resources, significant trade deficits, dependence on imports, limited know-how and human resource training, and very limited accessibility [2–6]. Many of these countries also face high environmental vulnerability and considerable risk from the impacts of climate change [7,8].

As a development strategy, most SIDSs have promoted the opening of their economies to foreign investment, accompanied by efforts in diversification and specialization in several products and services [4,9–11]. In this context, activities such as the exploitation of natural resources, including certain agricultural products, energy, and fisheries, as well as investments in the financial and tourism sectors, stand out [12]. Additionally, various SIDS have undertaken significant economic reforms, optimizing conditions for foreign investment, stimulating private sector growth, modernizing their governance systems, and strengthening regional cooperation [4,13].

The internationalization of the economy has been crucial to the economic development of many of the SIDS, but it has also introduced a series of new and complex challenges.

Various authors criticize this development model, arguing that it intensifies dependency on exports as well as exacerbates inequalities, and the main beneficiaries are large international corporations, thus perpetuating a dependency dynamic reminiscent of neocolonial patterns [5,14,15].

Research on Small Island Developing States (SIDSs) has provided a unique context for the study of development models [2,13,16–19]. Research on these territories has evolved from focusing on limitations and vulnerabilities to seeking opportunities within each reality and building resilience based on sustainability principles [10,12,18,20]. Despite the specificities of each SIDS, there are many commonalities. Among these, most researchers highlight tourism as a strategic sector for the development of these territories. According to the World Tourism Organization [21], tourism has become the main export category for many SIDSs, creating many jobs and development opportunities.

This research aims to contribute to sustainable tourism development on the island of Príncipe, part of the São Tomé and Príncipe archipelago, a SIDS located in the Gulf of Guinea. Due to its small size, economic weakness and high vulnerability, tourism development must not only ensure sustainability in the environmental, economic and socio-cultural dimensions but also promote the 17 Sustainable Development Goals outlined by the United Nations in the 2030 Agenda for Sustainable Development.

The island is an interesting case study, characterized by its double insularity and a unique tourism development model, targeted at high-income tourists who value sustainability and authenticity in destinations [22]. In addition to the fact that the tourism supply is dominated by international companies, this destination is in an early stage of development, offering a very limited range of tourism products and experiences.

Through the review of literature, characterization of the territory, conducting an inventory of resources, and applying interviews with various stakeholders, it was concluded that it is crucial to establish the foundations for a tourism model more oriented toward the community, aiming for more effective social and economic sustainability. In line with this objective, this study proposes the enhancement of the territory's tourism supply through the design of products and tourist experiences. To achieve this, a well-defined methodology for developing tourism products and experiences has been implemented, which is underpinned by a series of carefully structured stages [23–25]. The work was supported by extensive fieldwork, collaboration with various experts, and semi-structured interviews.

## 2. São Tomé and Príncipe: Geography and Tourism

### 2.1. Insularity and Economy

The Republic of São Tomé and Príncipe (STP) is a volcanic archipelago located in the Gulf of Guinea, spanning just 1001 km$^2$ and with a population of about 232,000 spread over two islands. Independent since 1975, this Portuguese-speaking country is currently a multi-party, semi-presidential democracy.

During the extensive colonial period, the island's economy was sustained by the production of sugar cane and later by monocultures of coffee and cocoa [26]. Throughout this period, a Creole society emerged, which was associated with the biological and cultural mixing of Portuguese settlers and African slaves [27]. The cultural identity and much of the intangible and material heritage from this colonial period are important assets that enrich the territory.

Although it is less than 300 kilometers from the African coast, the country experiences significant isolation and territorial dispersion, being heavily reliant on air transportation for external connections. This increases the cost of commercial transactions and limits travel.

Classified as lower–middle income, with a GDP per capita of about $2400, STP is a small island state with "a fragile economy highly vulnerable to exogenous shocks" [28]. Indeed, the country faces structural challenges common to many of the SIDSs. Its isolation and small scale are reflected in limited and undiversified productive activity.

The dependence on external energy, exacerbated by underdeveloped infrastructures and small scale, creates additional costs, further conditioning the business environment.

All this leads to a strong dependence on imports and a consequent high trade deficit. According to data from the IMF [29], estimates of the trade deficit in 2023 pointed to USD 132.4 million, representing 23% of GDP. Among the exports of goods, cocoa and palm oil stand out [30]. In turn, the tourism sector (an estimated USD 132.4 million in 2023), remittances from emigrants (around USD 10 million in 2022, equivalent to 1.9% of GDP) and international financial assistance (14.0% of GDP in 2022) are essential to mitigate the chronic deficit in the balance of payments [29].

The country is still marked by high poverty, pronounced income inequality (reflected in a Gini index of 40.7) and scarce employment opportunities [31,32]. According to the OECD [33], São Tomé and Príncipe is one of the central African countries with the highest unemployment rates between 2000 and 2015 with an average unemployment rate of 15% and a significantly higher rate of 22.6% among the youth.

The rapid population growth poses a challenge to its economy, but that approximately half of its population is under 18 years old can be seen as an opportunity, especially as they have a high level of education, with a secondary school enrollment rate of 89% [32].

Despite its economic limitations, the country possesses a set of strengths and opportunities, notably its natural wealth and beauty, high biodiversity, and excellent beaches. These characteristics, coupled with the uniqueness of its culture, a hospitable population, and a high level of safety, are favorable factors for tourism development.

### 2.2. Tourism in São Tomé and Príncipe

Tourism is considered a key and emerging sector in the country, contributing directly to 6 to 8% of the GDP, or 14% when its indirect impact is included [29,34]. It is the second-largest export sector after agriculture [31]. In 2019, the number of tourists visiting STP was around 35,000. After a sharp decrease in 2020 and 2021 due to the COVID-19 pandemic, tourist arrivals in 2023 are expected to exceed the 2019 figures [34].

Portugal stands out as the primary source market for tourists, accounting for over 50% of visitors. Some Portuguese tour operators provide regular packages, typically of 8 to 12 days, with prices starting at 1200 euros or slightly higher for extensions to Príncipe. Market diversification is essential for tourism growth but could increase the risk of undue pressure from tourists.

The Strategic and Marketing Plan for Tourism in São Tomé and Príncipe 2018–2025 sets forth a strategic vision for 2025 to position this SIDS as "the most preserved insular tourist destination in Equatorial Africa, with unique nature and biodiversity, paradisiacal beaches, a historical-cultural legacy of coffee and cocoa plantations, and a hospitable local community that enriches the tourist experience" [34]. The plan suggests that the current tourism development model, heavily reliant on external investment, should gradually evolve toward a community-based model to ensure greater sustainability and return for the local community.

### 2.3. Geography and Tourism on Príncipe Island

The island of Príncipe, about 160 km northeast of São Tomé, has been an autonomous region since 1995. With an area of 142 km$^2$, it has just over 8000 inhabitants, representing only 4% of the country's population (Table 1). Access to Príncipe is almost exclusively from São Tomé either by plane or boat. The number of flights has increased significantly in recent years, to around 16 per week, but aircraft have limited capacity, and fares are relatively high. There are sea connections, but they are not on a fixed schedule and only operate when the number of passengers and cargo justify the trip, making them impractical for international tourism.

**Table 1.** Area and population in São Tomé and Príncipe.

| | Area | | Population (2023) | | Population |
| | km$^2$ | % | No.º | % | Density |
|---|---|---|---|---|---|
| São Tomé | 859 | 85.8 | 222,750 | 96.1 | 259.3 |
| Príncipe | 142 | 14.2 | 9150 | 3.9 | 64.4 |
| STP | 1001 | 100 | 231,900 | 100 | 231.7 |

Source: [28,31,32].

Due to its isolation and very small market, the island faces significant development challenges. Its economy is quite precarious, largely based on informal services and trade, as well as agriculture and fishing, which are mainly subsistence based [34]. In recent decades, tourism has emerged as the driving force behind the island's transformation and development.

Although the population is predominantly young and educated, technical and higher education is very limited [21]. In addition, many young people leave the island after completing compulsory education.

The territory can be divided into two regions with distinct geological and climatic characteristics. The southern region, more mountainous and humid, contrasts with the northern region, which is less rainy and where most of the population resides. Santo António, despite having just over a thousand inhabitants, is the capital of the island and is where the main public services are concentrated.

With a hot and humid equatorial climate throughout the year, the seasons are defined by the distribution of rainfall. Due to its climate, geomorphology, and low population density, the island is covered with dense and lush vegetation. In 2012, the island was declared a UNESCO World Biosphere Reserve, and about 54% of the territory is classified as a nature park.

This natural wealth, combined with the beauty of its beaches, safety, authenticity, and the uniqueness of its culture, are key factors for tourism development. Both the central and local governments recognize tourism as the main strategic sector for the economy [34]. In 2019, about 8000 tourists visited the island, representing 23% of the tourists arriving in the country. Although many of the tourists who visit the country only stay on the island of São Tomé, when considering the density of demand (tourists/km$^2$) and the tourism saturation index (guests/population × 100), the figures for the island of Príncipe are significantly more significant than those for the island of São Tomé (Table 2).

**Table 2.** Tourism and resources in São Tomé and Príncipe.

| | Tourists 2019 | Demand Density | Tourist Saturation Index | Accommodation (2018) | | Restaurants 2019 |
| | | | | Units | Beds | |
|---|---|---|---|---|---|---|
| São Tomé | 27,000 | 31.4 | 12.5 | 51 | 1734 | 139 |
| Príncipe | 8000 | 56.3 | 96.4 | 19 | 224 | 11 |
| STP | 35,000 | 35.0 | 15.6 | 70 | 1958 | 150 |

Source: [32,34].

Despite the richness and uniqueness of its tourism resources, the development of tourism has been greatly hindered by a set of constraints that impede investment in infrastructure, accommodation units, and other services. The most important of these are poor accessibility, lack of skilled labor, scarcity of capital, lack of an entrepreneurial social fabric and high operating costs.

These constraints, along with the central government's limited involvement, explain why the island's tourism development, at least in the early stage, must be driven from outside. Currently, foreign companies dominate the entirety offer of the island's hotel accommodations and a substantial portion of other tourist services. These international investors have become the main catalysts for tourism activity on the island. In particular, the group Here Be Dragons (HBD) plays a major role in various initiatives aimed at the

sustainable development of the island. Its projects pay special attention to environmental conservation and social needs, and its approach has been instrumental in creating employment opportunities and boosting the island's economy. In addition to the four hotels owned by international groups, there are a few small accommodation facilities on the island, most of which are in Santo António.

The range of tourist experiences on the island is very limited with no specialized companies in this area. Many of the island's potential tourist products remain unexplored. The few existing experiences are mainly provided by the hotel units or by some associations or independent guides.

The tourism development of the island is guided by the UN-HABITAT's Príncipe 2030 strategic plan, aiming for sustainable development resilient to climate change [35]. Like many island territories, STP is highly vulnerable to the impacts of climate change, particularly sea level rise and the increasing frequency and intensity of storms [7]. The vision for tourism on Príncipe Island is to position the destination for a high-end market, targeting experienced travelers who prefer responsible travel that combines nature, sea, and culture [22,34].

## 3. Methodology

This research aims to foster sustainable tourism development on the island of Príncipe. The initial phase encompassed a comprehensive review of the pertinent literature, which was supplemented by conducting seven semi-structured interviews with key stakeholders at the destination (Table 3). Engagement with these experts aimed to provide a holistic understanding of the destination.

**Table 3.** Stakeholders and personalities interviewed.

| Stakeholder | Entity | Position | Interviewee |
|---|---|---|---|
| Main tourist agents | HBD Group | Activities Coordinator | André Rosa |
| | | Chief Sustainability Officer | Emma Tuzinkiewicz |
| | Hotel Belo Monte | Director | Miles Oats |
| Public institutions | Príncipe Island Guide Center | President | Yodiney dos Santos |
| | Príncipe Natural Park Biosphere Reserve | Head of Section Acting | Alberto Leal |
| | Department of Tourism and Commerce | Director | Giffrey Teixeira |
| ONGs | Principe Foundation | President | Estrela Matilde |

The interview script has 15 questions grouped into three dimensions: (i) characterization of the destination; (ii) stakeholder intervention in the tourism development process; and (iii) the potential for tourism development on the island. The interviews were conducted face-to-face in June 2022.

The results showed that to promote a development model that ensures greater participation by the local community, it is essential to expand the range of tourist experiences on offer; to this end, in the second phase of this study, the island's tourism resources were inventoried and categorized. A methodology for developing tourism products and experiences was adopted based on four phases [21,24].

The first consisted of defining the conceptual model, selecting the resources to be covered and the sources and stakeholders to be consulted. This phase is essential to ensure the quality of the resource inventory due to the complexity of the process and the specificities of each destination.

In the second phase, an inventory of tourism resources was conducted. This process involved systematically aggregating information on the tourist destination's tangible and intangible heritage. For effective organization, the collected data were catalogued in a georeferenced database, incorporating essential variables for each resource. The inventory was conducted using secondary sources, and extensive fieldwork was carried out throughout 2022.

The next phase was to assess the tourism potential of the resources. In addition to the knowledge gained from direct observation, collaboration was sought from eight selected experts, considering their relevance, and understanding of the products (Table 4). Through interviews, they contributed (i) to the assessment of the relevance of the inventory; (ii) to the identification of other resources; and (iii) to the classification of each resource, following the methodology suggested by the World Tourism Organization [21,25]. The evaluation of tourist resources was based on four criteria: attractiveness, uniqueness, potential to provide high emotional value experiences, and accessibility.

**Table 4.** Experts who collaborated in the evaluation of tourism products.

| Product | Entity/Position | Specialist Name |
|---------|-----------------|-----------------|
| Pedestrian routes | HBD Experience Manager | André Rosa |
| Extraction activities/Processing/Manual arts | Príncipe Natural Park—PNP | Júlio Mendes Daniel Ramos |
| Birdwatching | President of the Príncipe Guides Association Prince Foundation Coordinator | Yodiney dos Santos |
| Artisanal fishing | Directorate of fisheries | Damião Matos |
| Scuba dive | Director Hotel Belo Monte | Miles Oats |
| Cultural resources | Head of the department of the cultural center and historical archive | Filomena Pina |
| Natural resources | Príncipe Natural Park—PNP | Maria José |

The final phase of the project involved the design of tourism experiences, following stages of ideation, implementation, evaluation, feedback, and co-creation. Several tourist experiences were tested, first with HBD employees and later with tourists. Through direct observation and feedback, the products were improved and customized. Resources were strategically pooled to form products and experiences tailored to increasingly segmented markets. Owing to the extensive amount of information and materials produced, this article provides only a summarized overview of the results achieved in this phase.

## 4. Results

### 4.1. Interviews

The results of the interviews are presented considering the three dimensions into which the questionnaire is divided. Regarding the first dimension of the interviews—Characterization of tourism in Príncipe (Questions Q1 to Q7), the respondents believe that Príncipe is positioned for a medium–high market, targeting responsible tourists who value the uniqueness and exclusivity of the destination. Ecotourism is the island's trademark, particularly in terms of nature tourism complemented by beach and cultural tourism (Q1). However, there is a strong dependence on the Portuguese market, and it is essential to expand the demand from other European countries with higher purchasing power and motivation for nature tourism. Most tourists who arrive in Príncipe come on tour packages, often as an extension of a visit to São Tomé, staying two to five days on the island. Generally, these tourists stay in

hotels belonging to international groups, but there is a growing demand from independent travelers (Q1, Q2).

In identifying the strengths and key tourist attractions, interviewees emphasize the island's natural park, a variety of trails, pristine beaches, stunning waterfalls, historical plantations ('roças'), the hospitality and culture of the local community, safety, historical heritage, boat trips, and turtle watching (Q3, Q7).

When discussing the primary challenges to tourism development, they note significant hurdles in terms of the qualification of human resources, accessibility, unknown destinations, and the precariousness of the health system (Q4).

Regarding the level of offer of experiences and the difficulties of development, they point out that it is limited to the services provided by hotels and some local guides, who lack qualifications and organization. But they consider great potential for expansion and improvement (Q5). For the tourism sector's growth on the island, there is a consensus on expanding the variety of tourism products and experiences. However, they identify several difficulties such as limitations in infrastructure, financing issues, lack of entrepreneurial culture, limited demand, ineffective regulation, and gaps in training and education (Q6).

Regarding the second dimension of the interviews—Stakeholder intervention in the tourism development process (Q9 to Q11), interviewees emphasize that tourism development has been driven primarily by international private investors. The two main hotel groups on the island not only dominate a large part of the tourism supply but also promote and fund various community and conservation projects. The HBD group is the island's largest employer, and the Príncipe Foundation manages several projects that promote greater social and environmental sustainability while enhancing tourism activities. These include guide training, turtle watching, a waste recycling cooperative, and the production of Abade soap and jewelry made from Picão flip-flops (Q8, Q9).

In contrast, the public sector has been criticized for its inactivity and inefficiency. Public sector investment in infrastructure and essential services to support tourism has been very limited, and regulation and supervision are inadequate (Q11). The local community should also play a more active role, but this requires a greater focus on training and encouraging entrepreneurship (Q10).

Regarding the third dimension of the interviews—The potential for tourism development on the island (Q12, Q15), respondents consider it essential to strengthen the sustainable development of tourism in Príncipe. They emphasize the need to mitigate the negative impacts of tourism, such as increased consumption of fossil fuels, waste production, cost of living, dependence on imports, and pressure on ecosystems. They stress the urgency of ensuring biodiversity conservation, implementing effective regulation and oversight, promoting collaboration among different stakeholders, and investing in human resource training. Concurrently, they highlight the need to raise awareness among the local population and tourists to promote sustainability and drive a circular economy, reducing dependence on imports and encouraging local production (Q12, Q15).

Additionally, they emphasize the need to strengthen the promotion of the destination and expand the range of tourism products and experiences. In connection with natural resources, they highlight the improvement of hiking trails, wildlife and flora observation activities, and the development of nautical activities. In terms of cultural tourism, they suggest promoting cultural events, valuing traditional therapy practices and the interpretation of medicinal plants, as well as the revitalization of historical sites. The built heritage, such as forts, churches, lighthouses, and other colonial buildings, can be enhanced, as well as the tangible and intangible heritage associated with 'roças' and former colonial agricultural enterprises (Q13, Q14).

### 4.2. Inventory and Evaluation of Tourist Resources

A detailed inventory of resources was compiled using secondary sources, fieldwork, and the collaboration of eight experts. The information was organized into databases and resource sheets. Table 5 provides a list of the main primary resources.

**Table 5.** Summary of primary resource inventory of Príncipe Island.

| Natural Heritage | | | | | |
|---|---|---|---|---|---|
| Ocean beaches | 49 | Peaks and mountains | 36 | Natural recreational spaces | 9 |
| River beaches | 7 | Cliffs/Ravines | 6 | Diving spots | 12 |
| Permanent rivers | 8 | Forest areas | 5 | Waves with surf potential | 2 |
| Waterfalls | 8 | Natural parks | 1 | Bird-watching spots | 9 |
| Bays | 7 | Nature reserves | 7 | Turtle-watching spots | 2 |
| Islanders | 7 | Mangroves | 3 | Whale-watching spots | 6 |
| Cultural Heritage | | | Other Features | | |
| Artistic heritage | 10 | Monuments | 22 | Official walking routes | 9 |
| Ethnographic heritage | 6 | Museums | 2 | Tourist routes | 13 |
| Roças | 14 | Lighthouses | 8 | Events | 10 |

The inventory highlights the natural resources, particularly the beaches, waterfalls, and species of significant interest for observation, such as birds and turtles. There are also good diving spots and potential for whale watching.

The combination of low population density and modest tourist demand, alongside the designation of over half of the island as a natural park with sprawling equatorial forests, are significant tourism assets that enhance ecotourism. Restrictions on access to protected areas and limitations on infrastructure and accessibility value conservation, but they also simultaneously limit the potential use of many tourist resources.

Cultural resources, while more limited, are noteworthy for their unique characteristics. These include the 'roças', historical plantations hailing from the colonial era, various colonial-era buildings, a few cultural events, and a rich heritage rooted in the island's agricultural and fishing practices. These elements collectively offer a glimpse into the island's historical and cultural landscape.

Regarding secondary resources, a notable feature is the airport, equipped to handle small and medium-sized flights, along with a range of excellent quality accommodation options: (i) three small luxury lodging units from the HBO group, offering a total of 46 rooms; (ii) the Belo Monte Hotel, situated in a former 'roça', providing 18 rooms; and (iii) Quinta de Santa Rita with 16 rooms. In addition to these hotels, there are five small local accommodations and a wild camping site. As far as catering options, apart from those offered by the accommodation units, the selection is very limited.

In terms of transport on the island, visitors have the flexibility to use services provided by their accommodations or hire a car. Although there are no official car hire companies, an informal network of vehicle rentals is available. Given that most of the roads are unpaved, it is advisable to choose an off-road vehicle. Alternatively, some tourists choose to explore the island on foot, take advantage of hitchhiking opportunities, or make use of the readily available motorcycle taxi services.

Based on the insights of eight experts involved in assessing the island's tourism products and potential, it is clear that while there is considerable scope for tourism growth, overcoming numerous challenges is essential. Aligning with the development constraints identified earlier in the interview phase, these experts highlight critical limitations in several key areas: the skills of human resources, the adequacy of infrastructure and equipment, overall accessibility, and the capabilities of the healthcare and emergency services systems.

Currently, the range of activities and tourist experiences offered, mainly by hotel units, includes boat tours, snorkeling, bird watching, fishing, turtle watching, whale watching, guided 4 × 4 tours, and hiking. Tours are the most popular and widely sold tourist activity on the island, which are operated either by hotels or local guides. In addition to visits to the main attractions, it often includes a local gastronomic experience either in a restaurant or as a picnic.

Hiking trails are one of the most valuable tourist assets of the island. Exploring Príncipe on foot offers an unparalleled experience, where visitors can safely and peacefully traverse dense and pristine tropical forests safely and without any concern for violence or wildlife-related dangers. There are nine official Biosphere trails, but those located within

the natural park have restricted access or are limited to research and conservation activities. André Rosa, the HBD Experience Manager, notes that most of these trails need investments to ensure good visiting conditions. Beyond these official paths, there is also a broad network of trails used by locals, several of which have tourism potential (Figure 1). Among these, André Rosa highlights the trail leading to Maria Correia waterfall and the trail to the Papagaio River dam.

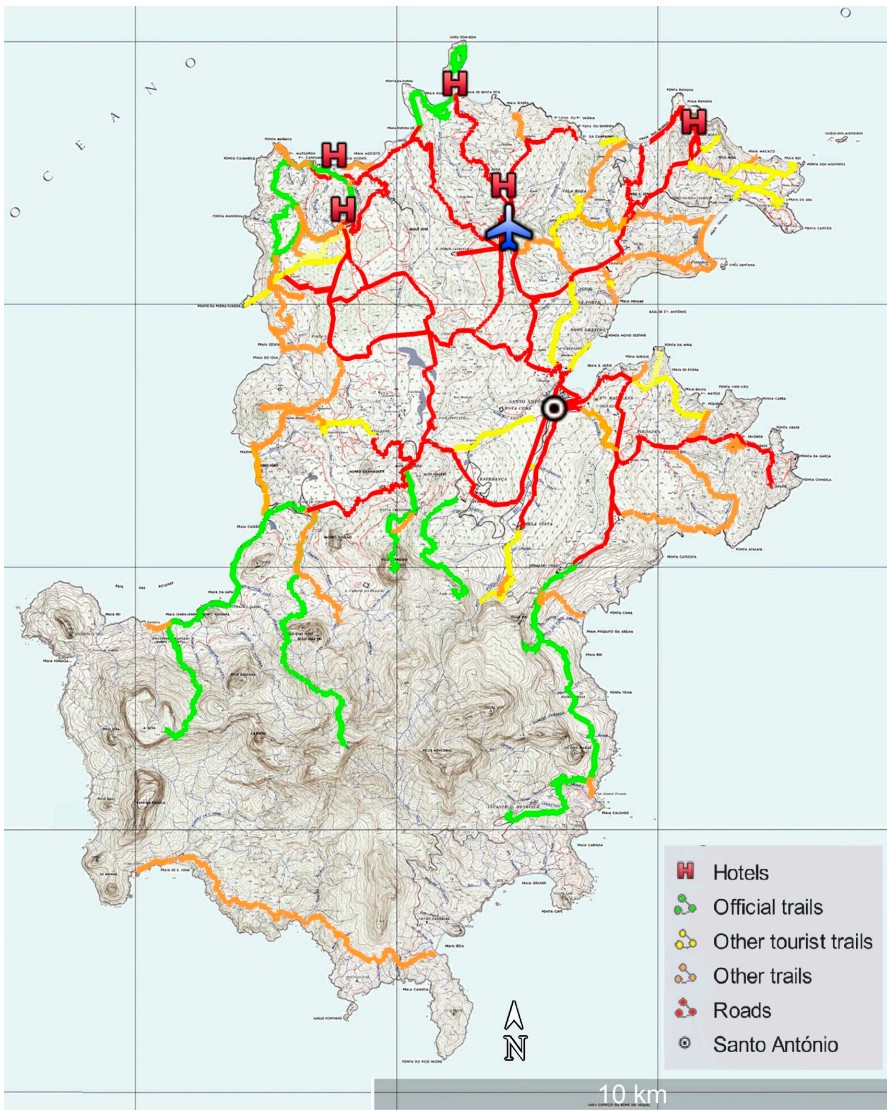

**Figure 1.** Pedestrian trails on Príncipe Island.

Another important tourist activity on the island is motorboat cruising. Sailing along the lush coast, with its many coves and paradisiacal beaches, in a sea of calm, blue waters, is an unforgettable experience. Boat trips also offer opportunities for snorkeling and observing birds and marine life. Most tourists engage in activities arranged by hotels, but it is also possible to hire the services of local fishermen, which adds a unique and authentic dimension to the experience.

Some activities are already well established in the tourism supply but should be expanded and improved. Turtle watching is a successful example of a sustainable activity with benefits for the conservation of the species and the involvement of the local community, which can serve as a reference for the design and management of other products in the region.

According to the Director of Fisheries of Príncipe (Damião Matos), an experience with development potential is to accompany the local community in their artisanal fishing activities. In addition to being possible all year round and offering a unique and authentic experience, this activity has the potential to create an alternative livelihood for fishermen and contribute to the sustainability of fisheries.

Experts also identify other activities such as scuba diving, yachting, and whale watching which, while having potential, require significant investments and may not be viable in the short term. Conversely, there is ample scope to develop more activities related to the observation of fauna and flora as well as those tied to agroforestry heritage. Notable among these are traditional practices linked to the cultivation of cocoa, coffee, vanilla, honey, palm oil, wine, and various medicinal plants. These activities, integral to ecotourism, provide educational and interactive experiences that engage with the local community and contribute to the destination's image of sustainability.

In addition, cultural interpretation activities are also noteworthy. Unique events such as the São Lourenço celebration, traditional dances and music, handicrafts, and the art of Andala basketry, as well as entrepreneurial initiatives linked to sustainability, like the Waste Valorization Cooperative, are highlighted as points of tourist interest.

*4.3. Summary of Empirical Investigation*

After a comprehensive analysis involving interviews, field surveys, and expert consultations, it is evident that Príncipe Island possesses considerable potential for tourism development. However, given the fragile nature of the territory and its various limitations, it is crucial to ensure that development is sustainable and brings greater involvement and benefits to the local community. In achieving this, action on three strategic axes is considered essential.

The first involves enhancing the range of tourism products and experiences, which should leverage the destination's primary resources, ensuring greater returns for the local community and sustainability.

The second axis entails a commitment to enhancing human resource skills and fostering local entrepreneurship. This strategy aims to elevate the community's qualifications and participation, upgrade service standards, and ensure the adoption of responsible tourism practices.

The third strategic axis focuses on governance and destination management. This involves reinforcing the role of public entities and community involvement, bolstering entrepreneurship, enhancing infrastructure, and implementing more transparent regulations. It also encompasses active promotion of the destination and adherence to a development model anchored in three key pillars: sustainability, social development, and quality. Effective implementation of these strategies hinges on robust governance characterized by transparency, commitment, and collaborative efforts among all stakeholders.

## 5. Conclusions

This research makes a significant contribution to the broader discourse on tourism development in SIDS and points the way forward for more sustainable and responsible tourism development in Príncipe. The Príncipe case study is particularly compelling in the field of tourism research in SIDS because of the sector's early stage of expansion and the uniqueness of its development model. Despite its reliance on international companies, the destination has maintained a commitment to small-scale operations, with branding strongly emphasizing authenticity and sustainability. The international hotel chains operating on the island have made significant investments in sustainability, actively supporting local projects focused on environmental conservation and contributing to social development.

However, the over-dependence on foreign investment and international companies hampers the future of more sustainable development, highlighting the need for a shift toward a more community-centered model.

Such a transition is essential to bring more tangible benefits to the local population and the country's economy and to enhance the tourist experience. But, considering the country's multiple constraints, this transition should be gradual, carefully exploring avenues for a harmonious coexistence with international investment.

The path toward a more community-centered model, while promising, is complex and requires a measured approach. This shift requires a multi-faceted investment: enhancing people's skills and qualifications, upgrading infrastructure, ensuring better governance, and focusing on small-scale ecotourism services. This holistic approach is key to raising the standard of tourist experiences and, importantly, to better the living conditions of the residents of Príncipe. By adopting this comprehensive approach, we can more effectively ensure a forward-looking, progressive path toward sustainable development.

The success of these strategies hinges on robust governance marked by transparency, dedication, and collaborative synergy among all stakeholders. By aligning these elements, Príncipe can advance its tourism sector and set a precedent for sustainable development that resonates across similar island economies.

The emphasis on local involvement and ownership is expected to produce more equitable economic growth with profits being reinvested in the community, thus promoting a more resilient and self-sufficient tourism sector. Furthermore, this model would encourage tourists to engage more deeply with the local culture and environment, potentially leading to a richer and more authentic travel experience.

However, this more community-centered model also carries risks and may not meet the expectations of some stakeholders who yearn for faster growth in tourism activity.

**Author Contributions:** Conceptualization, F.S.; methodology, F.S. and M.R.; investigation, M.R. and F.S.; writing—original draft preparation, F.S.; writing—review and editing, F.S. and M.R.; supervision, F.S. All authors have read and agreed to the published version of the manuscript.

**Funding:** This research received no external funding.

**Institutional Review Board Statement:** Not applicable.

**Informed Consent Statement:** Informed consent was obtained from all subjects involved in the study.

**Data Availability Statement:** Non-confidential data are available upon request.

**Acknowledgments:** The success of this article owes greatly to the invaluable cooperation and assistance provided by the community of Príncipe Island. Their active participation and support during our fieldwork were instrumental. We extend our heartfelt gratitude to the experts who generously contributed their time and insights for the interviews and evaluation of tourist resources, playing a pivotal role in enriching our study.

**Conflicts of Interest:** The authors declare no conflicts of interest.

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
