# Peer review of "Building the Framework for Sustainable Tourism in Príncipe Island"

_tourismhosp, doi:10.3390/tourhosp5010015_

Round 1
Reviewer 1 Report
Comments and Suggestions for Authors
It will be useful in the conclusions to make a bit more description on the " a more community-centred model" - now it is too wide understanding. To make it possible for the readers to realise the authors’ point of view.
The research addressed the up-to-date issue – correlation between demand for economy development – and tourism is a good source - and need to preserve the nature – from that point of view the tourism could be a proper activity as well in case there is a SD-approach.
It is not entirely clear whether the situation with tourism on the Príncipe Island is unique or whether it is typical for islands of this type. It would be good to compare it with a model territory - real or desired.
I would say, that the main added value is the characteristic of the concreate area – Príncipe Island — it is interesting to learn the situation with island tourism development.
It will be interesting to learn about the area in dynamic : what are the new organisaitons / institutions appeared in, let say, 5-10 years’ time, are there more institutes controlling the situation, has the situation gotten worse in line with more mass tourism? What indicators show that or what is the opinion of the experts involved by the author for interview?
There is not enough information about the availability and content of the program documents for tourism development (strategies, programs, concepts) and analysis of these documents from the perspective of solving problems identified during the interviews: are these problems known to the island administration and are they planned to be eliminated? How can the results obtained by the authors during the study be used to improve the situation with tourism and increase its sustainability?
Author Response
Thank you very much for your insightful feedback.
To strengthen the discussion on tourism development models, we've added a final sentence that emphasises the researchers' stance on the need for adaptation and acknowledges the potential for alternative perspectives.
This study has broadly assessed existing models without directly comparing them to specific islands, recognising the complex diversity that extends beyond the scope of Small Island Developing States (SIDS). Nevertheless, the suggestion of benchmarking against a theoretical or actual model area offers a compelling avenue for future research.
The article discussed businesses and other organisations involved in tourism in Príncipe; there aren't many, as the island is small. No significant changes are expected in the coming years, as accessibility and market access constraints prevent the development of mass tourism, and the current main players focus on small-scale tourism.
In addition, the study shows that the planning framework and government policies in São Tomé and Príncipe are predominantly geared to the needs and realities of the main island, São Tomé. However, our interviews underscored a deep recognition of the unique circumstances of Príncipe, which differ markedly from those of São Tomé. This research highlights the critical importance of recognising the diverse geographical and developmental contexts within SIDS, and argues for a nuanced approach that promotes both diversity and sustainability.
Reviewer 2 Report
Comments and Suggestions for Authors
The paper titled "Building the Framework for Sustainable Tourism in Príncipe Island" examines the challenges and opportunities for sustainable tourism development in Príncipe, a Small Island Developing State (SIDS) within the São Tomé and Príncipe archipelago. The study evaluates the suitability of the existing tourism development model, driven by international investment, and proposes a shift towards a community-centered approach to ensure greater social and economic sustainability. Through extensive fieldwork and stakeholder collaboration, the research emphasizes the need for a holistic strategy focusing on local skills enhancement, infrastructure improvement, governance enhancement, and diversification of tourism products and experiences. The paper outlines a methodology for developing tourism products and experiences, emphasizing community involvement and sustainability principles. It concludes by highlighting the importance of a gradual transition towards a more community-centered model to maximize benefits for the local population and enhance the overall tourist experience.
Comments:
1) How were the stakeholders identified and selected for the interviews conducted during the research?
2) Can you provide more details on the methodology used to inventory and categorize tourism resources on Príncipe Island?
3) What criteria were considered when assessing the tourism potential of the identified resources?
4) How were the experts involved in the evaluation of tourism products selected, and what specific roles did they play in the assessment process?
5) Can you elaborate on the stages involved in the design of tourism experiences, particularly regarding ideation, implementation, and co-creation?
6) What challenges were encountered during the research process, and how were they addressed?
7) How do you propose to measure the success and effectiveness of the proposed shift towards a community-centered tourism model?
8) What role do you envision for local governance structures in facilitating the transition towards sustainable tourism development?
9) How do you plan to disseminate the findings and recommendations of this research to relevant stakeholders and policymakers?
Author Response
Thank you for your insightful feedback.
Regarding your very pertinent comments, I respond as follows. Due to the limited number of characters in the article, only minor adjustments have been made.
- Because of its importance, one of the researchers spent 1.5 years on the island of Príncipe, which enabled the identification of key stakeholders.
- After six months of fieldwork, consulting both experts and secondary sources, several experiential products were developed and tested for HBD. In the final phase, the second researcher independently evaluated the products offered by tour operators and the local community.
- As added in the article, the evaluation was based on four criteria: attractiveness, uniqueness, potential to provide high emotional value experiences and accessibility.
- Their expertise in the territory and these products and their roles in design or management positions enabled this insight.
- The methodology included inventorying resources, rating them, clustering them, designing products, testing and evaluating them, and incorporating feedback.
- The lack of literature and official data on Príncipe was overcome through extensive fieldwork and stakeholder engagement.
- Reductions in unemployment and the Gini index, as well as an increase in local community provision of accommodation and other tourism services, are key indicators.
- Although Príncipe is an autonomous region, it lacks significant financial capacity, size and political clout. Some businesses and NGOs play a more important role in tourism development than the local government.
- We are committed to sharing our findings with all stakeholders. Following on from this study, we are now developing further research as part of a PhD, where we will incorporate this knowledge into a series of meetings (focus group methodology) with different stakeholders.
Reviewer 3 Report
Comments and Suggestions for Authors
Dear authors, I appreciated the article from a geographic point of view. A minor revision is needed to increase the article's impact on research.

Author Response
Thank you for your insightful feedback. A new version of the article has been submitted incorporating some changes suggested by the reviewers. In response to your very pertinent comments, I would like to highlight the following:
We agree that the study would be more robust with a wider range of interviewees and stakeholders. However, a second group of tourism product experts (n=8) were involved in practice, which added value to this study. Moreover, the researchers' presence on the island for several months ensured that the interviewees selected were the most relevant to our research.
The resource inventory and its evaluation included cross-referencing information from secondary sources, extensive fieldwork, and assessments by local experts.
Regarding the feedback and co-creation that were integrated into the design process, the following sentence was added: "Several tourist experiences were tested, first with HBD staff and later with tourists. Through direct observation and feedback, the products were improved and customized."
Round 2
Reviewer 1 Report
Comments and Suggestions for Authors
Now it looks better.
Reviewer 2 Report
Comments and Suggestions for Authors
I think the authors did a commendable job revising the manuscript. In response to my suggestions, the authors have modified the manuscript and I am confident it is ready for acceptance.
Reviewer 3 Report
Comments and Suggestions for Authors
Dear Authors, thanks for the revision, and now the manuscript is fine for publishing.
Best regards